# Is Fasting Superior to Continuous Caloric Restriction for Weight Loss and Metabolic Outcomes in Obese Adults? A Systematic Review and Meta-Analysis of Randomized Clinical Trials

**DOI:** 10.3390/nu16203533

**Published:** 2024-10-18

**Authors:** Víctor Siles-Guerrero, Jose M. Romero-Márquez, Rosa Natalia García-Pérez, Cristina Novo-Rodríguez, Juan Manuel Guardia-Baena, María Hayón-Ponce, Carmen Tenorio-Jiménez, Martín López-de-la-Torre-Casares, Araceli Muñoz-Garach

**Affiliations:** 1Department of Endocrinology and Nutrition, Virgen de las Nieves University Hospital, 18014 Granada, Spain; victor.siles.sspa@juntadeandalucia.es (V.S.-G.); rosan.garcia.sspa@juntadeandalucia.es (R.N.G.-P.); cristina.novo.sspa@juntadeandalucia.es (C.N.-R.); juanm.guardia.sspa@juntadeandalucia.es (J.M.G.-B.); maria.hayon.sspa@juntadeandaucia.es (M.H.-P.); carmen.tenorio.sspa@juntadeandalucia.es (C.T.-J.); martin.lopeztorre.sspa@juntadeandalucia.es (M.L.-d.-l.-T.-C.); 2Foundation for Biosanitary Research of Eastern Andalusia—Alejandro Otero (FIBAO), 18012 Granada, Spain; 3Granada Biosanitary Research Institute (Ibs. Granada), 18014 Granada, Spain; 4Physiopathology of Obesity and Nutrition Networking Biomedical Research Centre (CiberOBN), Carlos III Health Institute, 28029 Madrid, Spain

**Keywords:** intermittent fasting, daily caloric restriction, obesity, time restricted feeding, weight reduction, time restricted fasting, caloric deficit

## Abstract

Background: fasting-based strategies (FBS) and continuous caloric restriction (CCR) are popular methods for weight loss and improving metabolic health. FBS alternates between eating and fasting periods, while CCR reduces daily calorie intake consistently. Both aim to create a calorie deficit, but it is still uncertain as to which is more effective for short- and long-term weight and metabolic outcomes. Objectives: this systematic review and meta-analysis aimed to compare the effectiveness of FBS and CCR on these parameters in obese adults. Methods: after screening 342 articles, 10 randomized controlled trials (RCTs) with 623 participants were included. Results: both interventions led to weight loss, with a reduction of 5.5 to 6.5 kg observed at the six-month mark. However, the results showed that FBS led to slightly greater short-term reductions in body weight (−0.94 kg, *p* = 0.004) and fat mass (−1.08 kg, *p* = 0.0001) compared to CCR, although these differences are not clinically significant. Both interventions had similar effects on lean mass, waist and hip circumference, blood pressure, lipid profiles, and glucose metabolism. However, FBS improved insulin sensitivity, with significant reductions in fasting insulin (−7.46 pmol/L, *p* = 0.02) and HOMA-IR (−0.14, *p* = 0.02). Conclusions: despite these short-term benefits, FBS did not show superior long-term outcomes compared to CCR. Both strategies are effective for weight management, but more research is needed to explore the long-term clinical relevance of FBS in obese populations.

## 1. Introduction

Obesity is one of the most significant public health challenges globally, affecting over 890 million adults worldwide and contributing to the prevalence of chronic diseases such as type 2 diabetes, cardiovascular disease, hypertension, and certain cancers [1]. As a complex condition, obesity arises from the interplay of genetic, behavioral, environmental, and metabolic factors, leading to excess fat accumulation that poses severe health risks. The economic burden associated with obesity is also substantial, impacting healthcare systems through the treatment of obesity-related comorbidities and increasing indirect costs related to lost productivity and premature mortality [2].

Effective management of obesity is critical for improving overall health outcomes and reducing the burden on healthcare systems [2]. Among the various approaches to weight loss, dietary interventions play a pivotal role, with FBS and CCR as two prominent methods. Both interventions aim to reduce caloric intake, thereby inducing a caloric deficit essential for weight loss [3]. However, they differ fundamentally in their approach to achieving this deficit. Despite widespread use, there remains a lack of consensus on which strategy is more effective for long-term weight loss and metabolic health improvements.

The discussion surrounding FBS extends beyond a purely nutritional focus, touching on key issues such as long-term effectiveness, sustainability, and individual variability in metabolic responses. While evidence highlights the potential of FBS to enhance insulin sensitivity [4] and reduce inflammation [5], there are also concerns regarding possible nutrient deficiencies [6], the risk of promoting eating disorders [7], and potential hormonal or metabolic imbalances in susceptible populations [8]. Given these complexities, personalized recommendations for intermittent fasting, along with tailored nutritional interventions, are becoming increasingly recognized as essential. Individual differences in psychosocial environments, lifestyles, and health statuses necessitate customized approaches to maximize benefits while mitigating risks. This emphasizes the importance of addressing related complications through personalized dietary planning.

In this context, FBS have gained significant popularity in recent years, particularly due to their simplicity and the potential benefits they offer for both weight loss and metabolic health [9]. FBS encompasses a variety of eating patterns that alternate between periods of eating and fasting. The most common forms include intermittent fasting, time-restricted eating, and alternate-day fasting. In intermittent fasting, individuals alternate between fasting days and eating days or hours, such as the popular 5:2 method, where participants fast for two non-consecutive days and eat regularly for the remaining five. Time-restricted eating involves limiting the eating window to a specific number of hours per day, typically ranging from 4 to 12 h, with the remaining time spent fasting. Alternate-day fasting involves alternating days of normal eating with days where caloric intake is significantly reduced, usually by 25–30% of total daily energy expenditure [10]. These strategies have shown promise in various studies, with evidence suggesting that they may improve insulin sensitivity, reduce body fat, and enhance metabolic health, even independent of weight loss [3,9,10,11,12,13].

On the other hand, CCR remains the traditional and widely accepted dietary approach for weight loss. CCR involves reducing daily caloric intake by a specific percentage (typically 20–40%) without altering the frequency or timing of meals. This consistent reduction in energy intake has been the cornerstone of weight management programs for decades, with numerous studies supporting its efficacy in achieving weight loss and improving metabolic markers such as blood pressure, cholesterol levels, and glucose metabolism [14,15]. However, despite its proven effectiveness, adherence to CCR can be challenging for many individuals, as sustained caloric reduction often leads to feelings of deprivation, hunger, and eventual dietary relapse [16,17].

The effectiveness of FBS compared to CCR is still a subject of debate, with conflicting findings from various studies. This systematic review and meta-analysis aims to address the gaps in current research by comparing the effectiveness of FBS and CCR in terms of weight loss, body composition, and metabolic outcomes in obese adults. By synthesizing data from RCTs, this review seeks to provide a clearer understanding of whether FBS offers any distinct advantages over CCR for short- and long-term weight management.

## 2. Materials and Methods

This study was performed and prepared according to the guidelines proposed by Preferred Reporting Items for Systematic Reviews and Meta-Analyses (PRISMA) 2020 statement.

### 2.1. Search Strategy

In this systematic review with meta-analysis, two authors (V.S.-G. and J.M.R.-M.) systematically searched the PubMed database for RCTs up to 1 August 2024. The search string combined various terms related to intermittent fasting with terms related to weight loss in obese patients. Both Medical Subject Headings (MeSH) and free-text search terms were utilized. The search was restricted to human studies but did not include any limitations on the publication date. The following search strategy was employed: (“Intermittent fasting” OR “Time Restricted Feeding” OR “Time Restricted Feedings” OR “Time Restricted Fasting” OR “Time Restricted Eating” OR “Meal Skipping” OR “Breakfast Skipping”) AND (“weight loss” OR “Weight Losses” OR “Weight Reduction” OR “Weight Reductions”) AND (obesity OR obese) in the title or abstract.

### 2.2. In- and Exclusion Criteria

To select relevant studies, the following inclusion criteria were applied: (1) RCTs with a parallel group design; (2) RCTs involving participants from both intervention and control groups classified as obese (BMI ≥ 30 kg/m^2^); (3) RCTs where the control group underwent continuous calorie restriction; (4) participants aged 18 years or older; (5) studies that provided data on body weight changes (the primary outcome) before and after the intervention.

Exclusion criteria included the following: (1) studies without a properly controlled design for the intervention or control group, or those assessing other nutritional interventions; (2) in vitro studies, preclinical research, reviews, letters, comments, or any publications lacking primary data and/or clear methodology; (3) studies with intervention periods shorter than 4 weeks; (4) studies with insufficient baseline and final data for key outcomes in both groups; (5) studies including participants with acute or chronic conditions like gastrointestinal or renal diseases, or cancer, that could affect results; (6) studies that were unavailable after contacting corresponding authors. Discrepancies were resolved by consensus among the authors.

### 2.3. Outcomes

The primary outcome was the change in body weight between the fasting intervention groups and control groups. Secondary outcomes included changes in body composition (lean mass and fat mass), waist and hip circumference, blood pressure, lipid profiles (low-density lipoprotein cholesterol [LDL-C], high-density lipoprotein cholesterol [HDL-C], total cholesterol [TC], and triglycerides [TG]), and glucose metabolism (fasting blood glucose, hemoglobin A1c [HbA1c], homeostatic model assessment of insulin resistance [HOMA-IR], and fasting insulin) between the fasting intervention groups and caloric restriction control groups. The included studies were required to assess at least the primary outcome. Secondary analyses that involved repeated primary or secondary outcomes published in different articles were excluded by mutual agreement among the authors.

### 2.4. Data Extraction and Statistical Analysis

Based on the inclusion and exclusion criteria, two authors (V.S.-G. and J.M.R.-M.) sequentially enrolled the trials and extracted data independently into a predesigned database. The extracted information included publication details (such as the first author’s name and the year of publication), participant characteristics (age, sample size, and body weight), and both primary and secondary outcomes. Any discrepancies in eligibility between the two researchers were resolved through discussion. If needed, a third researcher (R.N.G.-P.) was involved to reach a consensus. To enhance the consistency of the results, RCTs were categorized based on intervention duration, with short-term interventions defined as those lasting up to 6 months and long-term interventions as those exceeding 6 months.

Data were analyzed using RevMan version 5.4.1 (Cochrane Collaboration, DerSimonian & Laird, 1986 [18]). The pooled effect sizes are presented as mean differences with 95% confidence intervals (CI), calculated using the mean and standard deviation (SD) values before and after the interventions. In cases where the SD was unavailable, it was derived from standard errors and CI for group means, following the approach outlined in the Cochrane Handbook for Systematic Reviews of Interventions [19]. For the trials that merely reported the data with figures, the online program Plot Digitizer (https://plotdigitizer.com, accessed on 23 August 2024) was used to estimate or extract data. Heterogeneity among the pooled studies was assessed using the *I*^2^ value, with *I*^2^ ≥ 50% indicating high heterogeneity. For cases where *I*^2^ was less than 50%, a fixed-effects model was employed; otherwise, a random-effects model was used. A meta-analysis was conducted for secondary outcomes when at least three independent results from different RCTs were available. Sensitivity analyses were performed by excluding one primary study at a time to assess the robustness and consistency of the findings. All statistical tests were two-sided, with a *p*-value of ≤0.05 considered statistically significant. The principal investigator reviewed all data extraction and analyses prior to the final evaluation, and any disagreements were resolved through panel discussions.

### 2.5. Bias Assessment

The quality of the studies was assessed using a modified version of the Cochrane Collaboration’s tool [20]. This tool examines the risk of bias in trials across several domains: random sequence generation, allocation concealment, blinding of participants and personnel, blinding of outcome assessment, incomplete outcome data, selective reporting, and other potential sources of bias. Each domain was classified as having a low, high, or unclear risk of bias.

## 3. Results

### 3.1. Study Selection and Characteristics

A total of 340 related articles were identified through PubMed and supplemented by two references identified through handsearching, resulting in a total of 342 publications for screening (Figure 1).

According to the exclusion criteria, 41 systematic reviews/meta-analyses, 114 generic reviews, 104 animal or in vitro studies, and 24 other communications were excluded. A total of 59 articles were then assessed for eligibility. Through careful screening of the full text of the articles, 49 records were excluded for the following reasons: inclusion of chronic disease patients in eight studies, lack of appropriate methodology or control group in twenty-eight articles, absence of the primary outcome in two studies, patients with BMI lower than 30 kg/m^2^ in five studies, one article concerning under-age patients, three articles concerning combinations with other interventions, and two inaccessible articles following a corresponding author request.

Ultimately, ten RCTs studies with 623 participants were included in the quantitative synthesis [21,22,23,24,25,26,27,28,29,30]. The main characteristics of the ten eligible studies are presented in Table 1.

### 3.2. Risk of Bias

The risk of bias of the studies are presented in Figure 2A,B, and Appendix A. Out of the ten studies evaluated, nine employed a specific method of random allocation [21,23,24,25,26,27,28,29,30], with the exception of Hutchinson et al. [22]. However, none of the studies provided sufficient details regarding allocation concealment, blinding of the research personnel, or data management practices [21,22,23,24,25,26,27,28,29,30]. All the RCTs reported their planned outcomes, with no evidence of selective reporting [21,22,23,24,25,26,27,28,29,30]. Finally, one study inadequately reported a high dropout rate (over 50%) and did not sufficiently detail the handling of missing data [21], while the other studies presented a low dropout rate [22,23,24,25,26,27,28,29,30].

Publication bias in studies examining the primary outcome of short- and long-term effects was assessed using funnel plots, as shown in Figure 2C. The symmetry observed in the funnel plots through visual inspection suggests the absence of publication bias. The sensitivity analysis results showed that excluding any single trial did not affect the overall findings.

### 3.3. Effect of Fasting Strategies on Body Weight

Nine out of the ten studies assessed focused on ponderal weight loss in obese patients who underwent interventions lasting less than six months [22,23,24,25,26,27,28,29,30]. The short-term effects of FBS on body weight is illustrated in Figure 3A. Both interventions led to weight loss, with a reduction of 5.5 to 6.5 kg observed at the six-month mark. However, the combined analysis of the nine studies (n = 565) revealed a statistically significant (*p* = 0.004) overall effect size (−0.94 kg [95% CI: −1.58, −0.31], indicating that FBS were more effective than CCR to reduce body weight. Although there was moderate heterogeneity (*I*^2^ = 42%), this variability was not statistically significant (*p* = 0.08).

Interestingly, when weight loss interventions are planned to extend beyond six months, the overall effects tend to disappear, as illustrated in Figure 3B. The results were consistent (*I*^2^ = 12%) across the five studies (n = 423), and the overall effect of the fasting-based interventions compared to CCR on body weight loss was not statistically significant (0.06 kg [95% CI: −0.98, 1.09], *p* = 0.34) [21,23,24,28,30].

### 3.4. Effect of Fasting Strategies on Body Composition, Waist and Hip Circunference

The meta-analysis examined the short-term effects of fasting strategies on body composition, specifically focusing on body lean mass, fat mass, waist circumference, and hip circumference (Table 2, Appendix A). A total of six studies involving 364 participants reported no significant effect on lean mass in the short term (*p* = 0.18) compared to the CCR group [22,23,24,26,27,30]. For body fat mass, data from eight studies (n = 453) revealed a statistically significant (*p* = 0.0001) reduction favoring the fasting group (−1.08 kg [95% CI: −1.63, −0.53]) [22,23,24,25,26,27,29,30]. However, the effects on waist circumference (n = 502) and hip circumference (n = 233) were not significantly different from the CCR group.

In terms of long-term effects, FBS did not significantly affect body composition, with three studies [23,24,30] reporting on lean mass (n = 253) and four studies [21,23,24,30] on fat mass (n = 311) compared to CCR. The long-term effects on waist circumference were similarly non-significant.

### 3.5. Effect of Fasting Strategies on Blood Pressure and Lipid Profiles

In the short term, data from six studies (n = 413) showed no significant effect of FBS on both systolic and diastolic blood pressure compared to CCR (Table 3, and Appendix A) [22,23,24,26,27,28]. Similarly, HDL-C and LDL-C levels, based on five studies with 378 participants, showed no significant differences between fasting strategies and CCR [22,23,24,26,28]. For total cholesterol, results from six studies (n = 421) revealed no significant change between the two interventions [22,23,24,25,26,28]. Lastly, the effect on triglyceride levels, analyzed from five studies with 378 participants, also failed to reach significance when comparing fasting strategies to CCR [22,23,24,26,28].

Regarding long-term effects, four studies involving 360 participants found no significant differences in systolic or diastolic blood pressure between fasting strategies and CCR [21,23,24,28]. Similarly, across three studies with 302 participants, there were no significant changes in HDL-C, LDL-C, or total cholesterol levels between the two interventions [23,24,28]. Long-term effects on triglycerides also showed no significant differences when comparing FBS to CCR [23,24,28].

### 3.6. Effect of Fasting Strategies on Glucose Metabolism

In the short term, data from six studies involving 421 participants showed no significant effect on fasting glucose levels compared to CCR (Table 4, Appendix A) [22,23,24,25,26,28]. However, fasting insulin levels, derived from four studies (n = 179), showed a significant reduction (*p* = 0.02), favoring the fasting strategies group [22,23,25,26]. In contrast, there was no significant effect on HbA1C levels, based on three studies involving 199 participants among groups [23,26,28]. Finally, the analysis of HOMA-IR from five studies (n = 209) revealed a modest but statistically significant (*p* = 0.04) reduction, favoring the FBS group (–0.14 [95% CI: –0.27, –0.01]) [22,23,24,25,26].

In terms of long-term effects, data from three studies [23,24,28] involving 311 participants indicated a slightly but significant (*p* = 0.00001) reduction in fasting glucose levels, with an mean difference of −3.59 mg/dL (95% CI: −3.92, −3.25), favoring the fasting group.

## 4. Discussion

Although the effectiveness of fasting-related strategies combined with continuous caloric restriction for weight loss has been previously reviewed in mixed populations of overweight and obese patients [31], this systematic review and meta-analysis provides a deeper exploration of the impact of FBS on both weight loss and metabolic outcomes specifically in a homogeneous population of obese patients. Additionally, it addresses previously unexplored aspects, such as the efficacy of intervention duration, body lean mass preservation, and the specific effects on LDL/HDL cholesterol levels, among others. The authors systematically searched and analyzed data from 10 eligible RCTs involving a total of 623 participants.

The findings suggest that FBS may lead to greater weight loss in obese participants compared to CCR. However, the mean difference in weight loss was modest, with a reduction of only 0.94 kg in interventions lasting less than six months. A similar pattern was observed with body fat loss, where short-term fasting interventions led to a greater reduction in body fat (−1.08 kg) compared to CCR. These results could be attributed to FBS, which often involve substantial caloric restriction greater than CCR, leading to immediate weight loss due to a significant reduction in calorie intake. This short-term effectiveness may be primarily attributed to the body’s response to a sudden decrease in available energy sources. Despite statistical significance, these results have limited clinical relevance. Effective weight management in obese patients typically requires interventions lasting at least one year to achieve meaningful reductions in body weight and fat mass [32,33,34]. Notwithstanding, several studies have shown that initial weight loss can boost motivation and adherence to weight loss programs, which is often associated with long-term success in weight loss [35,36,37]. However, the present research revealed that both continuous energy restriction and fasting strategies had a similar effect on body weight and fat reduction in interventions lasting longer than six months. FBS can be difficult to sustain long-term and may not foster lasting lifestyle changes. Compared to more gradual, sustainable weight management approaches, fasting may not offer superior long-term benefits, raising questions about its overall efficacy and practicality as a long-term solution. Therefore, these results should be considered within the context of their limited clinical applicability.

Some RCTs have analyzed the effect of fasting strategies on lean mas. Since fasting often leads to a negative energy balance and weight loss, comparisons between fasting and CCR in some systematic reviews suggest either similar [38] or improved [39] preservation of lean mass. The variation in these findings may be attributed to differences in the types of fasting or the self-selected meal frequency by participants [40]. In the current research, both short- and long-term interventions showed similar effects between FBS and CCR. A notable limitation of the studies evaluated is that many prescribe fasting strategies as the independent variable but do not explicitly control for dietary intake.

Similarly, the meta-analysis revealed comparable reductions in hip and waist circumference in both interventions, both in the short and long term. Hip and waist circumference are reliable surrogate markers for visceral fat mass, and their measurement is recommended for assessing obesity-related disease risk [41]. However, there is no universally accepted protocol for measuring circumferences. Numerous cross-sectional studies have attempted to identify which measurement site best reflects visceral adiposity and cardiometabolic outcomes across different populations. A systematic review of 120 studies evaluating various measurement criteria found that the choice of site did not affect morbidity and mortality outcomes [42]. However, these studies highlighted that the site of circumference measurement is crucial in assessing central obesity and cardiometabolic health, as significant variability in measurement techniques can impact the accuracy of estimations.

Regarding blood pressure and lipid profile, both interventions showed significant reductions in these cardiometabolic risk factors in both the short and long term, with no differences between the groups. It should be noted that CCR has been well-established as an effective tool for managing cardiometabolic risk factors in obese individuals [43,44] and even adults without obesity [45,46]. Therefore, these findings indicate that FBS did not provide additional benefits compared to continuous energy restriction.

Finally, glucose related metabolism was also analyzed. The short-term effects indicate that FBS may not have a substantial impact on fasting glucose and HbA1c levels further than CCR. These findings align with studies such as those by Mattson et al. (2017), which found that short-term fasting might not produce immediate improvements in glucose regulation but could have cumulative effects over time with CCR [47]. In contrast, fasting insulin levels showed a statistically significant reduction, indicating that fasting could lead to improvements in insulin sensitivity even in the short-term. The HOMA-IR also demonstrated a small but significant reduction, suggesting improved insulin sensitivity during short-term fasting in comparison with only continuous energy restrictions. This is supported by a recent meta-analysis performed by Yuan et al. (2022), which showed that fasting strategies may improve insulin sensitivity by reducing fasting insulin levels in obese patients [4].

In the long-term, FBS appears to have a more pronounced effect on glucose metabolism. Fasting glucose levels showed a significant reduction (−3.59 mg/dL) in comparison with CCR. This result aligns with a recent meta-analysis published in 2023, which suggests that fasting can serve as an effective preventative dietary strategy for pre-diabetic individuals as it supports long-term glucose control [48].

This analysis has some limitations. First, there was heterogeneity among the fasting interventions, with treatment duration likely to be a key source of variability. To address this, random effects models were applied, and sensitivity analyses were conducted to account for potential sources of heterogeneity. Second, many of the included RCTs were short-term interventions, lasting less than six months, which may limit insights into the long-term effects of fasting strategies on weight loss and metabolic outcomes. Furthermore, the studies did not consistently control dietary intake, which may have influenced the results. Finally, this systematic review with meta-analysis was conducted on studies involving a homogeneous population of obese patients with no reported metabolic or psychological comorbidities. Therefore, the results should be interpreted with caution before being applied to populations that are vulnerable to these conditions.

## 5. Conclusions

The present systematic review and meta-analysis revealed that fasting strategies led to slightly greater short-term reductions in body weight and fat mass compared to CCR, though these differences were not clinically significant (Figure 4). Both approaches had similar effects on lean mass and waist and hip circumference, as well as cardiometabolic markers such as blood pressure, lipid profiles, and glucose metabolism. However, fasting interventions were associated with improved insulin sensitivity, demonstrated by reductions in fasting insulin and HOMA-IR. Despite the short-term benefits, fasting strategies did not demonstrate superior long-term outcomes compared to CCR. Therefore, both methods appear equally effective for weight management, and further research is needed to explore the long-term clinical implications of fasting interventions in obese individuals.

## Figures and Tables

**Figure 1 nutrients-16-03533-f001:**
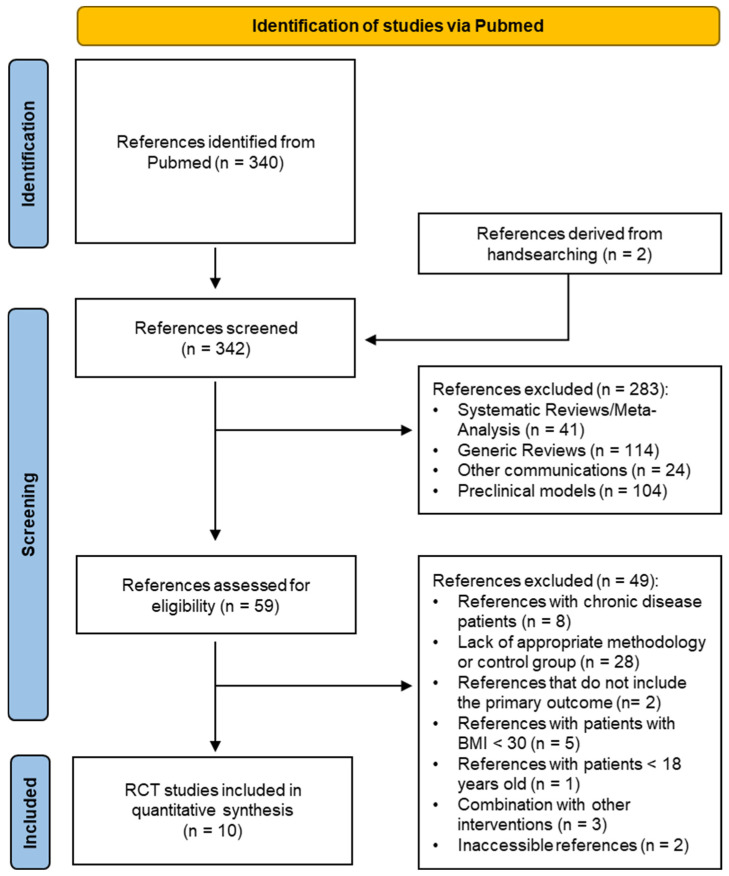
PRISMA flow diagram of study selection process.

**Figure 2 nutrients-16-03533-f002:**
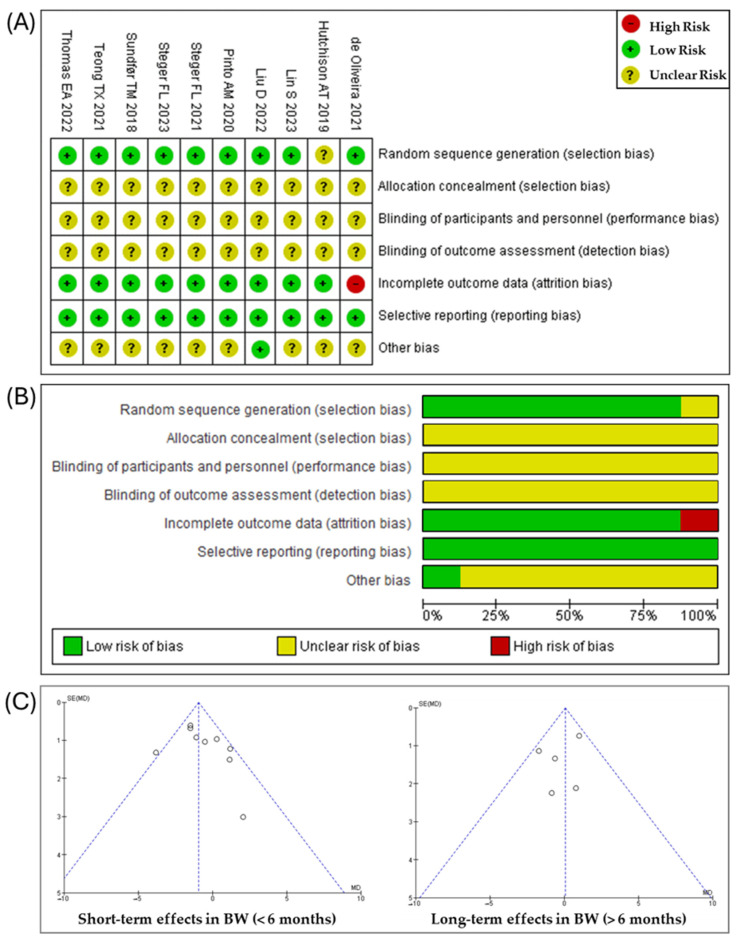
(**A**) Risk of bias summary: risk of bias item for each included RCT according to Cochrane Risk-of-Bias tool. (**B**) Risk of bias graph: each risk of bias item is presented as percentages across all included RCTs. (**C**) Funnel plot of publication bias for the primary outcomes of short- and long-term effects of this meta-analysis [21,22,23,24,25,26,27,28,29,30].

**Figure 3 nutrients-16-03533-f003:**
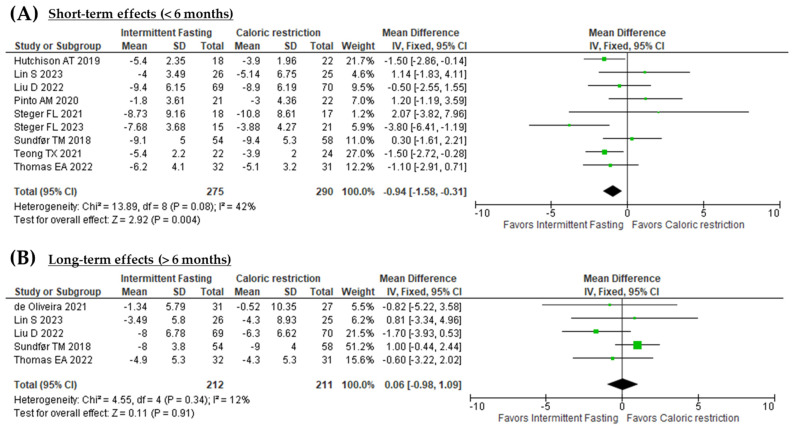
(**A**) Forest plot of the short-term effects (<6 months) of fasting based interventions versus continuous caloric restriction on body weight. (**B**) Forest plot of the long-term effects (>6 months) of fasting based interventions versus continuous caloric restriction on body weight [21,22,23,24,25,26,27,28,29,30].

**Figure 4 nutrients-16-03533-f004:**
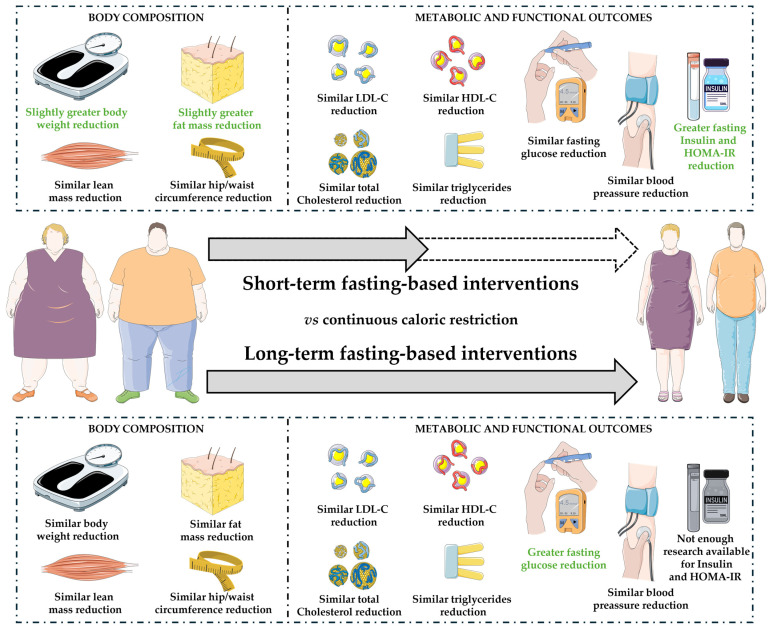
Representative illustration of the results obtained in the present meta-analysis on short- and long-term interventions based on fasting compared to continuous caloric restriction.

**Table 1 nutrients-16-03533-t001:** Basal characteristics of included studies.

	Sample (n; M/F)		Experimental Procedure			Population	
Studies	Int.	Control	Int.	Control	Duration	Basal Characteristics	Outcomes vs. Control
Lin S, 2023 [23]	30 (5/25)	30 (6/24)	TRE (8 h/16 h) + EW (12:00 p.m. to 08:00 p.m.)	25% CCR	24/48 weeks	Mid-aged (44 ± 12 years vs. 44 ± 9 years) obese (100 ± 17 kg bw vs. 102 ± 18 kg bw) patients	= BW, BLM, BFM, VFM, BP, WC, HDL, LDL, TC, TT, FG, HbA1c, HOMA-IR, and FI at 24/48 weeks
Steger FL, 2023 [26]	15 (4/11)	22 (6/15)	TRE (8 h/16 h) + EW (07:00 a.m. to 03:00 p.m.) + 25% CR	25% CCR	14 weeks	Mid-aged (46 ± 11 years vs. 42 ± 12 years) obese (111.1 ± 22.4 kg bw vs. 104.4 ± 21.7 kg bw) patients	↓ BW, BFM, FG, and HOMA-IR = FFM, VFM, WC, BP, TC, FI, Hb1Ac, LDL, and HDL
Liu D, 2022 [24]	69 (36/33)	70 (35/35)	TRE (8 h/16 h) + EW (08:00 a.m. to 04:00 p.m.) + 25% CR	25% CCR	24/48 weeks	Adult (31.6 ± 9.3 years vs. 32.2 ± 8.8 years) obese (88.4 ± 10.2 kg bw vs. 87.9 ± 12.8 kg bw) patients	= BW, WC, BFM, BLM, VFM, BP, FG, HDL, LDL, TC, TT, and HOMA-IR at 24/48 weeks
Thomas EA, 2022 [30]	41 (7/34)	40 (5/35)	TRE (10 h/14 h) + 35% CR	35% CCR	12/39 weeks	Adult (38.3 ± 7.9 years vs. 37.8 ± 7.8 years) obese (96.1 ± 18.1 kg bw vs. 93.4 ± 18.4 kg bw) patients	= BW, BFM, BLM, TC, TT, LDL, HDL, and HbA1c at 12/39 weeks
de Oliveira Maranhão Pureza IR, 2021 [21]	31 (0/31)	27 (0/27)	TRE (12 h/12 h) + 25% CR	25% CCR	48 weeks	Adult (31.80 ± 6.96 years vs. 31.03 ± 7.16 years) obese (81.25 ± 13.51 kg bw vs. 80.25 ± 9.40 kg bw) patients	↓ BFM at 48 weeks = BW, WC, and BP at 48 weeks
Steger FL, 2021 [27]	18 (5/13)	17 (3/14)	IF (3 non-CD of fasting (80% CR) + 4 days ad libitum)	25–35% CCR	12/24 weeks	Mid-aged (43.4 ± 11.8 years vs. 48.0 ± 10.0 years) obese (87.4 ± 11.5 kg bw vs. 91.0 ± 9.7 kg bw) patients	= BW, BFM, BLM, WC, HC, and BP at 12/24 weeks
Teong XT, 2021 [29]	22 (0/22)	24 (0/24)	IF (3 non-CD of fasting (70% CR) + 4 days of UE)	30% CCR	8 weeks	Mid-aged (50 ± 9 years vs. 50 ± 9 years) obese (89.2 ± 13.8 kg bw) patients	↓ BW and BFM = WC and HC
Pinto AM, 2020 [25]	21 (6/15)	22 (6/16)	IF (2 CD of fasting (80% CR) + 4 days of UE)	25% CCR	4 weeks	Mid-aged (50 ± 12 years vs. 56 ± 8 years) obese (87.7 ± 11.5 kg bw vs. 89.2 ± 13.8 kg bw) patients	↑ FG = BW, BFM, WC, FI, HOMA-IR, and HDL
Hutchison AT, 2019 [22]	22 (0/22)	24 (0/24)	IF (3 non-CD of fasting (70% CR) + 4 days of UE)	30% CCR	8 weeks	Mid-aged (49 ± 2 years vs. 51 ± 2 years) obese (89.4 ± 2.8 kg bw vs. 88.4 ± 2.8 kg bw) patients	↓ BW, BFM, WC, TC, LDL, and TG = BLM, HC, HDL, FG, FI, HOMA-IR, and BP
Sundfør TM, 2018 [28]	54 (28/26)	58 (28/30)	IF (2 non-CD of fasting (75% CR) + 5 days ad libitum)	25% CCR	12/24/48 weeks	Mid-aged (49.9 ± 10.1 years vs. 47.5 ± 11.6 years) obese (108.6 ± 16.3 kg bw vs. 107.5 ± 16.1 kg bw) patients	= BW, WC, HC, BP, HDL, LDL, TT, FG, and HbA1c at 12/24/48 weeks

Data are expressed as mean ± SD. Abbreviations: BLM, body lean mass; BFM, body fat mass; BP, blood pressure; BW, body weight; CD, consecutive days; CCR, continuous caloric restriction; CR, caloric restriction; EW, eating window; FG, fasting glucose; FI, fasting insulin; F, female; FFM, Free fat mass; HbA1c, glycated hemoglobin A1c; HC, hip circumference; HDL, high-density lipoprotein; HOMA-IR, homeostasis model assessment of insulin resistance; IF, intermittent fasting; LDL, low-density lipoprotein; M, male; TC, total cholesterol; TRE, time-restricted eating; TT, total triglycerides; UE, usual eating; VFM, visceral fat mass; WC, waist circumference. “=” means similar effect; ↓ means reduction; ↑ means higher.

**Table 2 nutrients-16-03533-t002:** Short- and long-term effects of fasting on body composition, waist and hip circumference.

Outcome	Studies	Sample	Statistical Method	Overall Effect Size	*p* Value
**Short-term effects**					
Body lean mass (kg)	6	364	MD (IV, Fixed, 95% CI)	−0.29 [−0.70, 0.13]	0.18
Body fat mass (kg)	8	453	MD (IV, Fixed, 95% CI)	−1.08 [−1.63, −0.53]	0.0001
Waist circumference (cm)	8	502	MD (IV, Random, 95% CI)	0.21 [−1.38, 1.80]	0.80
Hip circumference (cm)	4	233	MD (IV, Fixed, 95% CI)	−0.20 [−1.50, 1.10]	0.76
**Long-term effects**					
Body lean mass (kg)	3	253	MD (IV, Fixed, 95% CI)	0.07 [−0.45, 0.58]	0.80
Body fat mass (kg)	4	311	MD (IV, Fixed, 95% CI)	−0.96 [−2.09, 0.17]	0.10
Waist circumference (cm)	4	360	MD (IV, Random, 95% CI)	−0.90 [−3.12, 1.32]	0.42

Abbreviations: CI, confidence interval; IV, inverse variance; MD, mean difference.

**Table 3 nutrients-16-03533-t003:** Short- and long-term effects of fasting on blood pressure and lipid profiles.

Outcome	Studies	Sample	Statistical Method	Overall Effect Size	*p* Value
**Short-term effects**					
Systolic blood pressure (mmHg)	6	413	MD (IV, Fixed, 95% CI)	−0.80 [−2.89, 1.29]	0.45
Diastolic blood pressure (mmHg)	6	413	MD (IV, Fixed, 95% CI)	−1.03 [−2.44, 0.38]	0.15
HDL-Cholesterol (mg/dL)	5	378	MD (IV, Fixed, 95% CI)	−0.51 [−1.90, 0.89]	0.48
LDL-Cholesterol (mg/dL)	5	378	MD (IV, Random, 95% CI)	−3.19 [−10.61, 4.22]	0.40
Total cholesterol (mg/dL)	6	421	MD (IV, Random, 95% CI)	−3.15 [−8.84, 2.54]	0.28
Triglycerides (mg/dL)	5	378	MD (IV, Fixed, 95% CI)	−6.51 [−16.30, 3.28]	0.19
**Long-term effects**					
Systolic blood pressure (mmHg)	4	360	MD (IV, Fixed, 95% CI)	0.93 [−1.40, 3.27]	0.43
Diastolic blood pressure (mmHg)	4	360	MD (IV, Fixed, 95% CI)	−0.60 [−2.34, 1.13]	0.49
HDL-Cholesterol (mg/dL)	3	302	MD (IV, Fixed, 95% CI)	1.09 [−1.06, 3.24]	0.32
LDL-Cholesterol (mg/dL)	3	302	MD (IV, Fixed, 95% CI)	−1.33 [−6.66, 4.00]	0.62
Total cholesterol (mg/dL)	3	302	MD (IV, Fixed, 95% CI)	−0.62 [−6.63, 5.39]	0.84
Triglycerides (mg/dL)	3	302	MD (IV, Fixed, 95% CI)	−5.00 [−17.82, 7.83]	0.45

Abbreviations: CI, confidence interval; IV, inverse variance; MD, mean difference.

**Table 4 nutrients-16-03533-t004:** Short- and long-term effects of fasting on glucose metabolism.

Outcome	Studies	Sample	Statistical Method	Overall Effect Size	*p* Value
**Short-term effects**					
Fasting glucose (mg/dL)	6	421	MD (IV, Random, 95% CI)	−0.99 [−5.54, 3.56]	0.67
Fasting insulin (pmol/L)	4	179	MD (IV, Fixed, 95% CI)	−7.46 [−13.77, −1.15]	0.02
HbA1C (%)	3	199	MD (IV, Fixed, 95% CI)	−0.05 [−0.14, 0.05]	0.36
HOMA-IR	5	209	MD (IV, Fixed, 95% CI)	−0.14 [−0.27, −0.01]	0.04
**Long-term effects**					
Fasting glucose (mg/dL)	3	311	MD (IV, Fixed, 95% CI)	−3.59 [−3.92, −3.25]	0.00001

Abbreviations: CI, confidence interval; IV, inverse variance; MD, mean difference.

## Data Availability

The original contributions presented in the study are included in the article, further inquiries can be directed to the corresponding authors.

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
