# Peer review of "Is Fasting Superior to Continuous Caloric Restriction for Weight Loss and Metabolic Outcomes in Obese Adults? A Systematic Review and Meta-Analysis of Randomized Clinical Trials"

_nutrients, 2024, doi:10.3390/nu16203533_

Round 1
Reviewer 1 Report
Comments and Suggestions for Authors
This is a very well-executed systematic review and analysis on differences in weight loss and secondary clinical parameters between intermittent fasting and continuous calorie restriction. The quite clear outcome of the analysis is very well interpretated and discussed. I have just a few comments that might be helpful for increasing clarity and impact as follows:
1. As the authors define “Fasting-based strategies” (FBS) simply as strategies alternating between eating and fasting periods, I would suggest to simply use “Intermittent Fasting” or IF as abbreviation, instead of creating yet another uncommon abbreviation.
2. As in most other systematic reviews and meta-analyses, I always have the question is there something interesting in all the data that was excluded. In this case, the ratio of total to excluded studies is more than 30-fold for the first step and 6-fold for the second step. Although the authors are discussing the outcomes of some of the excluded studies, isn’t there a way to provide some idea of whether the bulk of excluded studies agrees or disagrees with the outcome of the 10 included studies? Specifically, some information about the outcome and general trend of the >100 preclinical models, which are often much better controlled, would be of great interest, at least to me.
3. Although I realize that the analysis was aimed at the difference, I would also like to know what kind of weight loss one can expect from each type of intervention. Glancing at Figure 3, it looks like that at 6 months the average (of the 9 studies) weight loss is ~ 6kg for both IF and CCR. I think it is important to provide this information even in the abstract, as it relativizes the small difference of ~ 1kg.
4. Finally, there are e few grammatical oddities, such as on line 222, “planned for extender over six months”; on line 79, should be “aims to”.
Comments on the Quality of English LanguageEnglish is fine
Author Response
This is a very well-executed systematic review and analysis on differences in weight loss and secondary clinical parameters between intermittent fasting and continuous calorie restriction. The quite clear outcome of the analysis is very well interpretated and discussed. I have just a few comments that might be helpful for increasing clarity and impact as follows:
AUTHORS: The authors appreciate the reviewer's positive and constructive comments, which will undoubtedly enhance the overall quality of the manuscript.
- As the authors define “Fasting-based strategies” (FBS) simply as strategies alternating between eating and fasting periods, I would suggest to simply use “Intermittent Fasting” or IF as abbreviation, instead of creating yet another uncommon abbreviation.
AUTHORS: The authors fully agree with the reviewer's suggestion regarding the use of abbreviations. However, during the discussion panel held prior to the manuscript's preparation, it was determined that the term "intermittent fasting" does not encompass all the specific features of other interventions, such as time-restricted eating or alternate-day fasting. Since fasting is the common element, the decision was made to group these approaches under the term "fasting-based strategies." Nonetheless, Table 1 identifies each specific type of fasting used in the studies, ensuring that each intervention is distinguished accordingly.
- As in most other systematic reviews and meta-analyses, I always have the question is there something interesting in all the data that was excluded. In this case, the ratio of total to excluded studies is more than 30-fold for the first step and 6-fold for the second step. Although the authors are discussing the outcomes of some of the excluded studies, isn’t there a way to provide some idea of whether the bulk of excluded studies agrees or disagrees with the outcome of the 10 included studies? Specifically, some information about the outcome and general trend of the >100 preclinical models, which are often much better controlled, would be of great interest, at least to me.
AUTHORS: The authors acknowledge that the inclusion criteria set by the working group led to the exclusion of numerous studies that did not employ a parallel randomized clinical trial (RCT) design in humans. As the reviewer highlights, preclinical trials often allow for greater control over experimental conditions, which is where clinical trials can face limitations. Interventions that are either unsustainable or cannot be fully assessed in real-world clinical practice tend to produce outcomes that differ significantly from those observed in tightly controlled preclinical settings. As Smith L. (2022) noted, preclinical studies frequently exhibit a positive bias, with almost all interventions yielding significant results (https://doi.org/10.3389%2Ffnbeh.2022.805661). This can overstate the effectiveness of certain interventions and distort their applicability in clinical practice. To mitigate these issues, the authors focused solely on parallel-design RCTs, ensuring that control groups were concurrent and drawn from distinct but comparable populations. This approach minimizes potential confounding factors, such as the effects of washout periods commonly used in other RCT designs, which can influence metabolism and weight loss outcomes.
- Although I realize that the analysis was aimed at the difference, I would also like to know what kind of weight loss one can expect from each type of intervention. Glancing at Figure 3, it looks like that at 6 months the average (of the 9 studies) weight loss is ~ 6kg for both IF and CCR. I think it is important to provide this information even in the abstract, as it relativizes the small difference of ~ 1kg.
AUTHORS: The authors appreciate the reviewer's comments. It is true that we have focused primarily on the differences between the interventions. To address this, a sentence has been added in the abstract and the main text to clarify that both interventions are effective in the short term, and the observed weight loss has been included for further clarification.
Abstract. Line 27 to 28: “Both interventions led to weight loss, with a reduction of 5.5 to 6.5 kg observed at the six-month mark. However, the results showed that FBS led to slightly greater short-term reductions in body weight (–0.94 kg, p = 0.004) and fat mass (–1.08 kg, p = 0.0001) compared to CCR, although these differences are not clinically significant.”
Main text. Line 225 to 226: “Both interventions led to weight loss, with a reduction of 5.5 to 6.5 kg observed at the six-month mark. However, the combined analysis of the nine studies (n = 565) revealed a statistically significant (p = 0.004) overall effect size (–0.94 kg [95% CI: –1.58, –0.31], indicating that FBS were more effective than CCR to reduce body weight”
- Finally, there are e few grammatical oddities, such as on line 222, “planned for extender over six months”; on line 79, should be “aims to”.
AUTHORS: The authors appreciate the indication of grammatical errors. We have corrected them in the main text.

Reviewer 2 Report
Comments and Suggestions for Authors
I think it is a great piece of work comparing these two popular strategies for weight loss. I only have one comment is that there was a recent very similar publication from the same journal. https://www.ncbi.nlm.nih.gov/pmc/articles/PMC9099935/pdf/nutrients-14-01781.pdf
The present authors need to remove the opening sentence to discussion that this is the first review, and also acknowledge the said review in 2022. It would be very helpful to highlight why this review is necessary and what are some of the flaws of the previous review, e.g., inclusion of pilot trials.
Author Response
I think it is a great piece of work comparing these two popular strategies for weight loss.
AUTHORS: The authors sincerely appreciate the reviewer's positive and constructive feedback, which will undoubtedly contribute to improving the overall quality of the manuscript.
I only have one comment is that there was a recent very similar publication from the same journal. https://www.ncbi.nlm.nih.gov/pmc/articles/PMC9099935/pdf/nutrients-14-01781.pdf
The present authors need to remove the opening sentence to discussion that this is the first review, and also acknowledge the said review in 2022. It would be very helpful to highlight why this review is necessary and what are some of the flaws of the previous review, e.g., inclusion of pilot trials.
AUTHORS: The authors thank the reviewer for their valuable clarification. A new paragraph has been added highlighting the novelty of this manuscript and its broader relevance to the scientific community. Line 293 to 299: “Although the effectiveness of fasting-related strategies combined with continuous caloric restriction for weight loss has been previously reviewed in mixed populations of overweight and obese patients [22], this systematic review and meta-analysis provides a deeper exploration of the impact of FBS on both weight loss and metabolic outcomes specifically in a homogeneous population of obese patients. Additionally, it addresses previously unexplored aspects, such as the efficacy of intervention duration, body lean mass preservation, and specific effects on LDL/HDL cholesterol levels, among others. Authors systematically searched and analyzed data from 10 eligible RCTs involving a total of 623 participants.”

Reviewer 3 Report
Comments and Suggestions for Authors
Is Fasting Superior to Continuous Caloric Restriction for Weight Loss and Metabolic Outcomes in Obese Adults? A Systematic Review and Meta-Analysis of Randomized Clinical Trials
Review
A brief summary
This systematic review and meta-analysis compared fasting-based strategies (FBS) and continuous caloric restriction (CCR) for weight loss and metabolic health in obese adults. After reviewing 342 articles, 10 randomized controlled trials involving 623 participants were included. Results indicated that FBS led to slightly greater short-term reductions in body weight (–0.94 kg) and fat mass (–1.08 kg) compared to CCR, though these differences weren't clinically significant. Both methods had similar effects on lean mass and metabolic markers, but FBS improved insulin sensitivity. However, FBS did not show superior long-term outcomes over CCR. Both strategies are effective for weight management, but further research is needed to assess the long-term benefits of FBS.
General concept comments
Article:
The results of this article are very interesting and well described.
Review:
v The Introduction section should be expanded to address the ongoing debate surrounding the FBS, not only in comparison to CCR. For example, reference number 3, titled "Effects of Different Types of Intermittent Fasting Interventions on Metabolic Health in Healthy Individuals (EDIF): A Randomised Trial with a Controlled-Run in Phase," specifically reported that LDL cholesterol levels increased significantly during the intervention phase for the 16/8 and 20/4 intermittent fasting cohorts, while LDL levels remained unchanged in the alternate-day fasting (ADF) group. Furthermore, the Introduction section should include a description of personalized recommendations for intermittent fasting and nutritional intervention approaches, as these may be crucial for addressing the associated complications. The following papers should be cited to support this discussion:
· Intermittent fasting and time-restricted eating role in dietary interventions and precision nutrition. https://doi.org/10.3389/fpubh.2022.1017254
· Understanding Type 2 Diabetes Mellitus Risk Parameters through Intermittent Fasting: A Machine Learning Approach. https://doi.org/10.3390/nu15183926
· The road ahead of dietary restriction on anti-aging: focusing on personalized nutrition. https://doi.org/10.1080/10408398.2022.2110034
Specific comments:
v In line 31 "However, FBS improved insulin sensitivity, as indicated by significant reductions in fasting insulin" should be replaced with "However, FBS improved insulin sensitivity, with significant reductions in fasting insulin" for smoother phrasing.
Comments on the Quality of English LanguageIn line 31 "However, FBS improved insulin sensitivity, as indicated by significant reductions in fasting insulin" should be replaced with "However, FBS improved insulin sensitivity, with significant reductions in fasting insulin" for smoother phrasing.
Author Response
A brief summary
This systematic review and meta-analysis compared fasting-based strategies (FBS) and continuous caloric restriction (CCR) for weight loss and metabolic health in obese adults. After reviewing 342 articles, 10 randomized controlled trials involving 623 participants were included. Results indicated that FBS led to slightly greater short-term reductions in body weight (–0.94 kg) and fat mass (–1.08 kg) compared to CCR, though these differences weren't clinically significant. Both methods had similar effects on lean mass and metabolic markers, but FBS improved insulin sensitivity. However, FBS did not show superior long-term outcomes over CCR. Both strategies are effective for weight management, but further research is needed to assess the long-term benefits of FBS.
General concept comments
Article:
The results of this article are very interesting and well described.
AUTHOR: The authors sincerely appreciate the reviewer's positive and constructive feedback, which has been carefully considered and incorporated to improve the quality and clarity of the manuscript. These valuable suggestions have strengthened the overall rigor and presentation of the work.
Review:
v The Introduction section should be expanded to address the ongoing debate surrounding the FBS, not only in comparison to CCR. For example, reference number 3, titled "Effects of Different Types of Intermittent Fasting Interventions on Metabolic Health in Healthy Individuals (EDIF): A Randomised Trial with a Controlled-Run in Phase," specifically reported that LDL cholesterol levels increased significantly during the intervention phase for the 16/8 and 20/4 intermittent fasting cohorts, while LDL levels remained unchanged in the alternate-day fasting (ADF) group. Furthermore, the Introduction section should include a description of personalized recommendations for intermittent fasting and nutritional intervention approaches, as these may be crucial for addressing the associated complications.
AUTHORS: The authors appreciate the comments provided in the introduction, as they offer valuable context that will enhance the overall manuscript. Line 56 to 66: The discussion surrounding FBS extends beyond a purely nutritional focus, touching on key issues such as long-term effectiveness, sustainability, and individual variability in metabolic responses. While evidence highlights the potential of FBS to enhance insulin sensitivity [4] and reduce inflammation [5], there are also concerns regarding possible nutrient deficiencies [6], the risk of promoting eating disorders [7], and potential hormonal or metabolic imbalances [8]. Given these complexities, personalized recommendations for intermittent fasting, along with tailored nutritional interventions, are becoming increasingly recognized as essential. Individual differences in psychosocial environments, lifestyles, and health statuses necessitate customized approaches to maximize benefits while mitigating risks. This emphasizes the importance of addressing related complications through personalized dietary planning.
The following papers should be cited to support this discussion:
- Intermittent fasting and time-restricted eating role in dietary interventions and precision nutrition. https://doi.org/10.3389/fpubh.2022.1017254
- Understanding Type 2 Diabetes Mellitus Risk Parameters through Intermittent Fasting: A Machine Learning Approach. https://doi.org/10.3390/nu15183926
- The road ahead of dietary restriction on anti-aging: focusing on personalized nutrition. https://doi.org/10.1080/10408398.2022.2110034
AUTHORS: The authors appreciate the references provided by the reviewer and were added to the main text to improve the overall manuscript.
Specific comments:
v In line 31 "However, FBS improved insulin sensitivity, as indicated by significant reductions in fasting insulin" should be replaced with "However, FBS improved insulin sensitivity, with significant reductions in fasting insulin" for smoother phrasing.
AUTHORS: The authors appreciate the indication of grammatical improvement. We have corrected them in the main text.
